# First evaluation of the GEMS glyoxal products against TROPOMI and ground-based measurements

Eunjo S. Ha[1], Rokjin J. Park[1], Hyeong-Ahn Kwon[2], Gitaek T. Lee[1], Sieun D. Lee[1], Seunga Shin[1], Dong-Won Lee[3], Hyunkee Hong[3], Christophe Lerot[4*], Isabelle De Smedt[4], Thomas Danckaert[4], Francois Hendrick[4], and Hitoshi Irie[5]

[1]School of Earth and Environmental Science, Seoul National University, Seoul, Republic of Korea
[2]Department of Environmental & Energy Engineering, University of Suwon, Hwaseong-si, Gyeonggi-do, Republic of Korea
[3]National Institute of Environmental Research, Incheon, Republic of Korea
[4]Royal Belgian Institute for Space Aeronomy (BIRA-IASB), Brussels, Belgium
[5]Department of Earth Sciences, Chiba University, Chiba, Japan
*Now at Constellr, Brussels, Belgium

*Correspondence to*: Rokjin J. Park (rjpark@snu.ac.kr) and Hyeong-Ahn Kwon (hakwon@suwon.ac.kr)

**Abstract.** The Geostationary Environment Monitoring Spectrometer (GEMS) aboard the GEO-KOMPSAT-2B satellite is the first geostationary satellite launched to monitor the environment. GEMS conducts hourly measurements during the day over East and Southeast Asia. This work presents glyoxal (CHOCHO) vertical column densities (VCDs) retrieved from GEMS, with optimal settings for glyoxal retrieval based on sensitivity tests involving reference spectrum sampling and fitting window selection. We evaluated GEMS glyoxal VCDs by comparing them to TROPOMI and MAX-DOAS ground-based observations. On average, GEMS and TROPOMI VCDs show a spatial correlation coefficient of 0.63, increasing to 0.87 for Northeast Asia. While GEMS and TROPOMI demonstrate similar monthly variations in the Indochinese peninsula regions (R > 0.67), variations differ in other areas. Specifically, GEMS VCDs are higher in the winter and either lower or comparable to TROPOMI and MAX-DOAS VCDs in the summer across Northeast Asia. We attributed the discrepancies in the monthly variation to a polluted reference spectrum and high $NO_2$ concentrations. When we correct GEMS glyoxal VCDs as a function of $NO_2$ SCDs, the monthly correlation coefficients substantially increase from 0.16–0.40 to 0.45–0.72 in high $NO_2$ regions. When averaged hourly, GEMS and MAX-DOAS VCDs exhibit similar diurnal variations, especially at stations in Japan (Chiba, Kasuga, and Fukue).

## 1 Introduction

Glyoxal (CHOCHO) is the smallest di-carbonyl compound with a short atmospheric lifetime of fewer than 3 hours during daylight hours (Volkamer et al., 2006). Some glyoxal is directly emitted through biomass burning and biofuel utilization; however, the majority is produced via the oxidation of non-methane volatile organic compounds (NMVOCs) (Fu et al., 2008). Glyoxal is predominantly removed from the atmosphere by photolysis and reaction with OH radicals

(Volkamer et al., 2005). When glyoxal oxidizes in the presence of nitrogen oxides, it contributes to the secondary formation of ozone. Furthermore, the high solubility of glyoxal facilitates its absorption by aqueous aerosols and cloud droplets, forming secondary organic aerosols (SOAs) (Fu et al., 2008; Lerot et al., 2021). Given that ozone and SOAs are harmful air pollutants and agents of climate change, comprehending their VOC precursors is critical for managing air quality and climate.

The number of VOCs detectable from space is limited compared to the numerous VOCs existing in the atmosphere. Glyoxal and formaldehyde (HCHO) are examples of non-methane volatile organic compounds (NMVOCs) that are retrieved using ultraviolet (UV) to visible wavelengths. These compounds, predominantly produced by the oxidation of other VOCs and characterized by short atmospheric lifetimes, provide valuable insights into local VOC emissions when measured. Satellite observations offer a comprehensive overview for estimating top-down emissions due to their extensive spatial
coverage compared to ground-based, in situ measurements (Choi et al., 2022). The different yields of formaldehyde and glyoxal from NMVOCs add additional information to constrain individual NMVOC emissions. For example, formaldehyde is typically produced in large amounts from alkenes, while glyoxal is a high-yield product of aromatic compounds (Cao et al., 2018; Chan Miller et al., 2016; Dufour et al., 2009). Chan Miller et al. (2017) noted that while formaldehyde and glyoxal data are closely linked, the precise measurements of glyoxal could provide additional information, especially in
environments with low nitrogen oxides (NOx). Furthermore, due to the shorter atmospheric lifetime of glyoxal compared to formaldehyde, elevated concentrations of glyoxal are indicative of the immediate vicinity of wildfires and areas with intense photochemical reactions (Alvarado et al., 2020; Vrekoussis et al., 2010). The ratio of glyoxal to formaldehyde (RGF = [CHOCHO]/[HCHO]) has been used in several studies to differentiate the origins of VOC emissions, distinguishing between anthropogenic or biogenic sources (Digangi et al., 2012; Vrekoussis et al., 2010).

The observation of glyoxal from a sun-synchronous satellite was conducted by the SCanning Imaging Absorption spectroMeter for Atmospheric CHartographY (SCIAMACHY) instrument, which was launched in 2002 (Wittrock et al., 2006). This instrument has a pixel size of 60 km along the track and 120 km across the track, enabling global coverage in six days. Building upon SCIAMACHY's achievements, glyoxal columns with improved spatial and temporal resolutions have been retrieved from the Global Ozone Monitoring Experiment-2 (GOME-2) (Lerot et al., 2010; Vrekoussis et al., 2009) and
Ozone Monitoring Instrument (OMI)  (Alvarado et al., 2014; Chan Miller et al., 2014). The spatial resolutions for GOME-2 and OMI glyoxal data are $80 \times 40$ km² and $13 \times 24$ km², respectively, offering global coverage in 1.5 days and one day. Glyoxal columns retrieved from the TROPOspheric Monitoring Instrument (TROPOMI) exhibit the highest spatial resolution, at $3.5 \times 5.5$ km², with an overpass of 13:30 local time (Lerot et al., 2021). These Low Earth Orbit (LEO) satellite instruments have significantly contributed to mapping the spatial distribution of glyoxal globally. However, they are limited
in their ability to capture the diurnal variations of glyoxal, which are crucial for understanding its emissions, transport, and chemical reactions.

To address the limitations of sun-synchronous satellites, the Geostationary Environment Monitoring Spectrometer (GEMS) was launched aboard the GEO-KOMPSAT-2B satellite in February 2020 (Kim et al., 2020), providing trace gas

and aerosol measurements as the first geostationary satellite. GEMS performs hourly measurements across East and Southeast Asia, including parts of India, ranging from 6–10 times a day depending on the season. This study presents the retrieval of glyoxal data from GEMS using an algorithm that Kwon et al. (2019) developed for formaldehyde retrieval. The adaptation of this algorithm for glyoxal retrieval is detailed in Sect. 2. We examine the retrieval uncertainty in Sect. 3. We evaluate the GEMS glyoxal product by comparing it with TROPOMI data in Sect. 4, and GEMS glyoxal products are validated against ground-based observations in Sect. 5.

## 2 Description of the GEMS glyoxal algorithm

For glyoxal retrievals, we use the same retrieval algorithm for formaldehyde for GEMS, and detailed descriptions of the algorithm are explained in Kwon et al. (2019) and Lee et al. (2024). Here, we only focus on the distinctive features of glyoxal retrievals. The algorithm descriptions and the evaluation results in this paper are based on GEMS glyoxal V2.0, which has been the operational product since 2023. The GEMS system attributes and parameters for radiance fitting are summarized in Table 1. Retrieving GEMS glyoxal vertical column densities (VCDs) involves three steps. First, a radiative transfer equation is fitted to back-scattered radiances within a glyoxal's spectral absorption range. This spectral fitting process yields a glyoxal slant column density (SCD), representing the integrated concentration along the mean photon path. Subsequently, the SCD is converted to the VCD by dividing by the air mass factor (AMF). AMF converts SCD to VCD by accounting for the light path varying with viewing geometry, the atmospheric scattering from clouds, and the vertical profile of glyoxal. Lastly, background correction is performed by adding simulated concentration over the reference sector.

Glyoxal is a weak absorber within its absorption range compared to ozone and nitrogen dioxide, and the amount of glyoxal in the atmosphere is relatively low. Therefore, strong absorbers and instrument noise can significantly constrain glyoxal retrievals. The native spatial resolution of GEMS is $3.5 \times 8$ km$^2$, with nitrogen dioxide, ozone, formaldehyde, and aerosol products retrieved at this resolution. For the weaker absorbers, we use co-added products, including radiance, irradiance, surface reflectance, and cloud products. Specifically, we co-add products with 4 (2×2) and 16 (4×4) GEMS pixels to retrieve sulfur dioxide and glyoxal, respectively, to enhance the signal-to-noise ratio. This approach reduces the spatial resolution to about $14 \times 32$ km² for glyoxal over Seoul, South Korea, but results in the stable spectral fitting to obtain glyoxal SCDs.

### 2.1 Spectral fitting

The spectral fitting yields glyoxal SCDs by fitting the modeled radiative transfer equation to measured radiances. The modeled radiative transfer equation demonstrates the attenuation of the reference spectrum by gas absorptions based on the Lambert-Beer law (Kwon et al., 2019) and can be expressed as follows:

$$I(\lambda) = \left[ \left( aI_0(\lambda) + c_r\sigma_r(\lambda) \right) e^{-\sum_i SCD_i\sigma_i(\lambda)} + c_{cm}\sigma_{cm}(\lambda) \right] P_{sc}(\lambda) + P_{bl}(\lambda). \qquad (1)$$

In Eq. (1), $a$ represents an amplification factor applied to the reference spectrum $I_0(\lambda)$; $c_r\sigma_r(\lambda)$ accounts for the contribution of the Ring effect; the term $e^{-\sum_i SCD_i\sigma_i(\lambda)}$ characterizes the attenuation of light by absorbing species $i$; $c_{cm}\sigma_{cm}(\lambda)$ reflects the influence of the common mode; and $P_{sc}(\lambda)$ and $P_{bl}(\lambda)$ stand for scaling and baseline polynomials, respectively. Solar irradiance is commonly used as the reference spectrum in the UV to visible wavelengths. However, using solar irradiance to retrieve weak absorbers could result in systematic biases caused by spectral interference or instrumental limitations (Lerot et al., 2021). Therefore, we use measured radiances as the reference spectrum in the spectral fitting. We obtain the reference spectrum by averaging radiances from clean pixels for the past three days for each track and scene in the reference sector (120–150° E). This reference sector contains open oceans where interference of liquid water absorption occurs, which could lead to a high bias of glyoxal VCDs on land. However, we inevitably selected this region as a reference sector since it generally shows low concentrations of glyoxal and other pollutants.

The spectral fitting accounts for absorption by chemical species, including CHOCHO, $NO_2$, $O_3$, $O_4$, $H_2O$ (liquid), and $H_2O$ (vapor). In addition, the GEMS instrument's polarization sensitivity is included as a pseudo-absorber since GEMS is not equipped with a polarization scrambler. The polarization sensitivity values measured before the launch of the GEMS are depicted in Figure 5 of Lee et al. (2024). The polarization sensitivity values at the fitting window of glyoxal are incorporated for the spectral fitting. We use the most updated absorption cross-sections available as of now. $NO_2$ absorption cross-sections at two temperatures (220, 294 K) are used, considering the strong influence of $NO_2$ at the glyoxal absorption range (Table 1).

Glyoxal retrieval is highly sensitive to the selection of the fitting window due to its low optical depth in the atmosphere (Alvarado et al., 2014). We conduct sensitivity tests of fitting window selection to minimize fitting RMS and column uncertainty of the retrieved glyoxal averaged over the entire domain by varying lower and upper wavelengths with 0.5 nm increments. Figure 1 shows the results of our sensitivity tests and our optimal fitting window of 433.0–461.5 nm for glyoxal retrieval. The fitting window of 433.0–461.5 nm was selected considering its low fitting RMS and column uncertainty. However, we find that the differential slant column densities (dSCDs) over the reference sector (120–150° E) retrieved with this fitting window have a positive value, which could result in a high systematic bias, which we discuss below.

Figures 2a and 2b show glyoxal SCDs and root-mean-square values of spectral fitting residuals (fitting RMS) retrieved using radiance references. The 1st and 99th percentiles and the average of fitting RMS in August 2020 are $3.6 \times 10^{-4}$, $8.0 \times 10^{-4}$, and $5.6 \times 10^{-4}$, respectively. The fitting RMS are large over the Tibetan plateau and the southwestern part of the domain despite low VCDs, indicating the low credibility of the retrieval. Figure 2c shows one case of fitted optical depth and fitting residuals in Indonesia (15 August 2020; 0.6° N, 123.9° E). The fitting residuals oscillate centered along the optical depth, indicating that fitting residuals have no specific features.

**Table 1. Summary of operational GEMS system attributes and parameters for radiance fitting.**

| | | |
|---|---|---|
| GEMS system attributes | Spectral range | 300 – 500 nm |
| | Spectral resolution | < 0.6 nm |
| | Wavelength sampling | < 0.2 nm |
| | Field of regard (FOR) | ≥ 5000 (N/S) km × 5000 (E/W) km (5° S–45° N, 75–145° E) |
| | Spatial resolution (at Seoul) | < 14 km × 32 km for glyoxal (4 × 4 co-added pixels) |
| | Duty cycle | 6 ~ 10 times per day (six times in winter, ten times in summer) |
| | Imaging time | ≤ 30 min |
| Radiance fitting parameters | Fitting window (calibration window)[1] | 433.0–461.5 nm (431.3–463.5 nm) |
| | Reference | Three days average of measured radiances from easternmost swaths (120–150° E) under clear-sky conditions (cloud fraction < 0.4) |
| | Solar reference spectrum | Chance and Kurucz (2010) |
| | Absorption cross-sections | CHOCHO at 296 K (Volkamer et al., 2005) $O_3$ at 223 K (Serdyuchenko et al., 2014) $NO_2$ at 220 K and 294 K (Vandaele et al., 1998) $O_4$ at 293 K (Finkenzeller and Volkamer, 2022) $H_2O$ (vapor) at 283 K (Gordon et al., 2022) $H_2O$ (liquid) at 296 K (Mason et al., 2016) |
| | Ring effect | Chance and Kurucz (2010) |
| | Common mode | Online common mode from easternmost swaths (120–150° E) for a day |
| | Polarization correction | Polarization sensitivity vector at the central pixel of charge-coupled-device (CCD) |
| | Scaling and baseline polynomials | Third order |

[1]. The calibration window refers to the wavelength range used for the calibration. During calibration, a convolved solar reference spectrum is fitted to the measured irradiances and radiances to obtain wavelength shift parameters and polynomial parameters (Kwon et al., 2019).

## 2.2 Air mass factor

In the presence of atmospheric scattering, AMF can be formulated in terms of scattering weight ($w_z$) and vertical shape factor ($S_z$) (Palmer et al., 2001):

$$AMF = \int_0^\infty w_z S_z dz. \qquad (2)$$

Scattering weight is a function of the solar zenith angle, viewing zenith angle, relative azimuth angle, surface reflectance, cloud pressure, and cloud fraction. We use different values of these parameters for each latitude, longitude, and month. A look-up table of the scattering weight at 448 nm is constructed using VLIDORT v2.6 (Spurr, 2006). Surface reflectance is obtained from OMI Lambertian Equivalent Reflectance (LER) Climatology products (Kleipool, 2010), and cloud pressure and fraction are obtained from GEMS L2 cloud products.

The vertical shape factor is calculated using a global 3-D chemical transport model (GEOS-Chem v13.0.0) with 47 vertical layers and 0.25°×0.3125° horizontal resolutions in Asia (Bey et al., 2001; Wang et al., 2004). The KORUS v5 inventory was used for anthropogenic emissions (Woo et al., 2020). Biogenic emissions are taken from MEGANv2.1 (Guenther et al., 2012), and biomass burning emissions are taken from the monthly GFED4 inventory (van der Werf et al., 2010). We use monthly mean hourly vertical profiles from August 2020 to July 2021 to better represent diurnal variations.

## 2.3 Background correction

Glyoxal SCDs retrieved from the spectral fitting using the radiance reference are dSCDs that do not include background columns over the clean reference sector (120–150° E). Therefore, we use simulated vertical columns in the reference sector for background correction:

$$VCD(i,j) = \frac{SCD(i,j)}{AMF(i,j)} = \frac{dSCD(i,j) + AMF_0(lat)VCD_m(lat)}{AMF(i,j)}. \qquad (3)$$

AMF($i,j$) indicates the AMF at the $i$th cross-track (north-south direction) and $j$th along-track (east-west direction) positions, and AMF$_0$(lat) denotes the AMF over the reference sector. VCD$_m$ is simulated monthly mean hourly VCD zonally averaged in the reference sector (120–150° E) from the model used to construct AMF. Figure 3 shows glyoxal VCDs with and without background correction. The difference is large in the high latitudes where the reference sector is close to polluted sources. However, the background contribution shown in Figure 3c is lower than the offset value ($10^{14}$ molecules cm$^{-2}$) used for the background correction of the TROPOMI (Lerot et al., 2021) and SCIAMACHY (Wittrock, 2006) glyoxal column. The offset value of $10^{14}$ molecules cm$^{-2}$ is selected from the ship-based measurement over the Pacific Ocean (Sinreich et al., 2010). As the background contribution for GEMS glyoxal VCD is low, the VCDs with (Figure 3a) and without background correction (Figure 3b) do not represent significant differences. The low value of VCD$_m$ is due to the underestimation of glyoxal

columns from the current chemical transport models (CTM) (Li et al., 2018; Silva et al., 2018). Simulated glyoxal VCDs from the ECHAM/MESSy Atmospheric Chemistry (EMAC) model also showed lower values than airborne measurements from the EMeRGe-Asia campaign observing the East China Sea in early spring 2018 (Kluge et al., 2023). Previous studies

suggested that the emissions of precursor VOCs are underestimated (Choi et al., 2022; Kwon et al., 2021) and that the oxidative chemistry producing glyoxal is not well represented (Silva et al., 2018).

## 3 Uncertainty analysis

In this section, we examine the uncertainty of each retrieval step. Assuming that each retrieval step is uncorrelated,

the total uncertainty in the retrieval can be expressed as the sum of the uncertainties of each retrieval step (Boersma et al., 2004; Kwon et al., 2019; De Smedt et al., 2018):

$$\sigma_{VCD}{}^2 = \left(\frac{\partial VCD}{\partial dSCD}\sigma_{SCD}\right)^2 + \left(\frac{\partial VCD}{\partial AMF}\sigma_{AMF}\right)^2 + \left(\frac{\partial VCD}{\partial VCD_m}\sigma_{VCD_m}\right)^2 + \left(\frac{\partial VCD}{\partial AMF_0}\sigma_{AMF_0}\right)^2, \quad (4)$$

where $\sigma$ denotes uncertainty in each parameter. Using the relationship shown in Eq. (3), we can derive the sensitivity of VCDs to each parameter, resulting in the conversion of Eq. (4) as follows:

$$\sigma_{VCD}{}^2 = \frac{1}{AMF^2}\sigma_{SCD}{}^2 + \frac{SCD^2}{AMF^4}\sigma_{AMF}{}^2 + \frac{AMF_0{}^2}{AMF^2}\sigma_{VCD_m}{}^2 + \frac{VCD_m{}^2}{AMF^2}\sigma_{AMF_0}{}^2. \quad (5)$$

Uncertainty in each parameter includes both random and systematic components. Unlike systematic uncertainty, random uncertainty decreases for a spatial or temporal averaging in the ratio of $\frac{1}{\sqrt{N}}$, where N is the number of pixels averaged. In this study, we only consider random uncertainty in slant columns as random uncertainty in AMF and $VCD_m$ are difficult to separate from the systematic uncertainty in practice.

### 3.1 Uncertainties in slant columns

### 3.1.1 Random uncertainties and observation noise in slant columns

Random uncertainties in slant columns ($\sigma_{SCD,rand}$) are fitting uncertainties mainly resulting from instrument noise and can be calculated using Eq. (15) of Kwon et al. (2019):

$$\sigma^2_{SCD,rand,j} = RMS^2 \frac{m}{m-n} C_{j,j} C_{j,j}. \quad (6)$$

RMS is the root-mean-square value of fitting residuals, m and n are the number of spectral grids and fitting parameters, $C_{j,j}$ are diagonal components of a covariance matrix, and j is the subscript for fitting parameters. The 1st to the 99th percentiles of random uncertainties are $3.6 \times 10^{14}$ and $1.6 \times 10^{15}$ molecules cm$^{-2}$ in August 2020, with a mean of $8.6 \times 10^{14}$ molecules cm$^{-2}$. The random uncertainties of GEMS are higher than TROPOMI, as the 1st to the 99th percentiles of TROPOMI random

uncertainties are $4.4 \times 10^{14}$ and $1.0 \times 10^{15}$ molecules cm$^{-2}$ in August 2020 with a mean of $6.5 \times 10^{14}$ molecules cm$^{-2}$ in the GEMS field of regards (FOR).

The observation noise is large compared to the actual signal for glyoxal retrieval, and the credibility of glyoxal retrieval is known to be low in oceans due to low concentration and interference with liquid water absorption (Alvarado et al., 2014). Figure 4 illustrates VCDs over the Pacific Ocean and the observation noise estimated from its standard deviation. We followed the analysis from Lerot et al. (2021) to estimate the noise level of GEMS VCDs and compared them with TROPOMI VCDs. While Lerot et al. (2021) analyzed VCDs over 180–120° W, we analyzed GEMS VCDs over 130–146° E due to the limited coverage of the geostationary satellite. We filter out pixels over land with cloud cover (cloud fraction > 0.4) for the analysis. VCDs binned in 5° latitude bands range from $0.9 \times 10^{14}$ to $2.1 \times 10^{14}$ molecules cm$^{-2}$, which is notably lower than the scatter for all data. GEMS glyoxal observation noise ranges from $3.7 \times 10^{14}$ to $7.0 \times 10^{14}$ molecules cm$^{-2}$, comparable to the noise of TROPOMI VCDs.

### 3.1.2 Systematic uncertainties in slant columns

Systematic uncertainties in slant columns are associated with various factors such as wavelength calibration, bandpass function, residual stray light, and absorption cross-sections. We assessed the uncertainty resulting from absorption cross-section, as it is a preeminent factor contributing to the uncertainty in slant columns (Lerot et al., 2021; De Smedt et al., 2018). The uncertainty related to absorption cross-section ($\sigma_{SCD,abs}{}^2$) is estimated by matrix analysis, following Rodgers formalism (Rodgers, 2000):

$$\sigma_{SCD,abs}{}^2 = \sum_{j=1}^{n} dSCD_j{}^2 [G S_{bj} G], \quad (7)$$

where $j$ $(=1,...,n)$ is the subscript for absorbing species; $G$ is the matrix formed by $G = [K^T K]^{-1} K^T$, where $K$ is the matrix constructed from the absorption cross-sections with the dimension of m×n (m is the number of spectral grids); $S_{bj}$ (n×n) is the cross-section error covariance matrix. The diagonal components of $S_{bj}$ are assigned based on the uncertainties for the absorption cross-sections of each species: glyoxal (3%) (Volkamer et al., 2005), $NO_2$ (3%) (Vandaele et al., 1998), $O_3$ (2.6%) (Serdyuchenko et al., 2014), gas phase $H_2O$ (4%) (Gordon et al., 2022), and $O_4$ (3%) (Finkenzeller and Volkamer, 2022). The contribution of liquid water absorption is excluded in the calculation, as it had higher contributions than $2 \times 10^{15}$ molecules cm$^{-2}$ for some latitudes. The reference sector for glyoxal is mostly ocean, which represents high concentrations of liquid water. Using this reference sector for liquid water retrieval results in negative dSCDs with large absolute values over land, which could lead to a large contribution from the liquid water absorption cross-section. The dSCDs for each species are retrieved for March 2021.

For geostationary satellites, a radiance reference is required for every along-track, but the observation area is limited to certain regions such as Asia, North America, and Europe. As a result, the reference sector inevitably includes

polluted regions, and the selection of the reference sector significantly influences retrieval results. Therefore, we examine the systematic uncertainty arising from the current reference sector. Since October 2020, scan areas have been fixed to Half East (HE), Half Korea (HK), Full Central (FC), and Full West (FW). The easternmost longitude of HE, HK, FC, and FW scan areas are 152° E, 142° E, 142° E, and 133° E, respectively. Therefore, relatively more clean areas in the Pacific Ocean are not observed in the HK, FC, and FW scan areas. However, during the in-orbit test period (IOT) from August to September

2020, observations were frequently taken as Nominal daily scan, which covers 90–150° E (Figure 1 of Kwon et al., 2019). This enables us to obtain the reference spectrum over 120–150° E during the IOT. Figure 5 demonstrates the sensitivity of retrieved glyoxal VCDs to the selection of reference spectrum depending on the scan areas in August 2020. Figure 5a illustrates glyoxal VCDs retrieved with the reference spectrum taken from the reference sector of 120–150° E in August 2020. Figure 5b shows the same retrieved glyoxal VCDs but with the reference spectrum averaged over 120–133° E, which

is the narrow reference sector limited by the FW scan, frequently conducted after the IOT. VCDs in Figure 5b are about 22% lower on average than those in Figure 5a, possibly due to the effect of local pollution on the reference spectrum. We assumed the systematic uncertainty associated with the reference spectrum as the standard deviation of the difference in SCD retrieved from the reference sector of 120–133° E and 120–150° E averaged for the observations in August 2020.

Figure 6a shows the total systematic uncertainty in SCD and the contribution from each uncertainty source relative

to glyoxal SCD. The contribution from the absorption cross-section of species $j$ is calculated as $\sqrt{dSCD_j{}^2[GS_{bj}G]}$. For most latitudes, uncertainty associated with the reference sector represented the largest ratio, followed by the uncertainties associated with $NO_2$ and $O_3$ absorption cross-sections. Figure 6b illustrates the contribution of SCD, AMF, and background correction to the uncertainty in VCD. The uncertainty in VCD is dominated by the uncertainty in SCD.

**3.2 Uncertainty in AMF**

We estimate the uncertainty in AMF by the composite of uncertainties in surface albedo ($\alpha_s$), cloud top pressure ($p_c$), effective cloud fraction ($f_c$), and profile height parameter ($p_h$) (Kwon et al., 2019):

$$\sigma_{AMF}{}^2 = \left(\frac{\partial AMF}{\partial \alpha_s}\sigma_{\alpha_s}\right)^2 + \left(\frac{\partial AMF}{\partial p_c}\sigma_{p_c}\right)^2 + \left(\frac{\partial AMF}{\partial f_c}\sigma_{f_c}\right)^2 + \left(\frac{\partial AMF}{\partial p_h}\sigma_{p_h}\right)^2, \quad (8)$$

where $p_h$ is defined as the height below which 75% of glyoxal VCDs exist from the surface. The uncertainties $\sigma_{\alpha_s}$, $\sigma_{p_c}$, and

240 $\sigma_{f_c}$ are assigned as 0.02, 50hPa, and 0.05, respectively (De Smedt et al., 2018; Kwon et al., 2019). The uncertainty $\sigma_{p_h}$ is calculated as a standard deviation of $p_h$ of a priori profile, ranging from 68hPa to 293hPa for different latitudes. The sensitivities of each parameter to AMF are obtained from the look-up table of the scattering weight at 448 nm. Uncertainties in AMF contribute 7–14% to the uncertainties in VCD across the latitudes.

## 3.3 Uncertainty in background correction

The contribution of background correction to the total uncertainty in VCD can be expressed as the sum of the third and fourth terms on the right-hand side of Eq. (5). We assign the uncertainty in simulated background concentration ($\sigma_{VCD_m}$) as the standard deviation of simulated concentration for each latitude. $\sigma_{VCD_m}$ ranges from $2.5 \times 10^{12}$ to $1.9 \times 10^{13}$ molecules cm$^{-2}$, and $VCD_m$ ranges from $2.5 \times 10^{12}$ to $2.6 \times 10^{13}$ molecules cm$^{-2}$. The uncertainties in background correction represent lower values than those in SCD and AMF for most latitudes. The uncertainties in background correction represent relatively high values in the latitudes lower than 0° N and higher than 28° N, where the terrestrial sources influence the reference sector. The total systematic uncertainties in VCD range from 33% to 61%, which is comparable to the 30% to 70% uncertainty range in TROPOMI glyoxal VCDs (Lerot et al., 2021).

## 4 Comparison with TROPOMI data

We evaluate GEMS glyoxal retrieval by comparing GEMS glyoxal VCDs with TROPOMI from August 2020 to July 2021. For comparing GEMS and TROPOMI products, GEMS VCDs are averaged in 0.5° x 0.5° grids weighted by the overlapping area between pixels and grid boxes. For comparison, we used hourly GEMS data (FinalAlgorithmFlags = 0, cloud fraction < 0.4) at the TROPOMI overpass time in each region. TROPOMI L3 glyoxal data with a spatial resolution of 0.05° x 0.05° are regridded into the same 0.5° x 0.5° grids for comparison.

Figures 8a and 8b show GEMS and TROPOMI glyoxal VCDs averaged from August 2020 to July 2021 at 11:30–15:30 local time. We find good consistencies between the two products over the whole domain, with a correlation coefficient of 0.63 and a regression slope of 1.26 (Figure 8d). Both products show high VCDs in the Indochinese Peninsula and populated cities such as Shanghai and Guangdong due to biogenic or anthropogenic emissions of VOCs. However, GEMS is slightly higher than TROPOMI in Northeast Asia. This discrepancy may be partly due to the elevated NO$_2$ concentration. Lerot et al. (2021) conducted an empirical correction for strong NO$_2$ absorption for TROPOMI, which decreased glyoxal concentrations as a function of NO$_2$ SCDs. The correction of GEMS glyoxal VCDs accounting for the strong absorption NO$_2$ needs to be developed as the current operational product (GEMS glyoxal V2.0) does not consider this effect.

In the west of the domain, the negative bias of GEMS VCDs compared to TROPOMI occurs due to the GEMS's high viewing zenith angle (Figure 8c). A similar negative bias is also found in formaldehyde retrieval (Lee et al., 2024). GEMS glyoxal VCDs are even negative in parts of the Indian Ocean, the area outside the dashed green line depicted in Figure 7, in which no clear reasons for this issue have been found yet. Excluding this region in our scatter increases the correlation coefficient to 0.71 between the two products (Figure 8e). We also find an excellent agreement (R=0.87) between the two products in Northeast Asia, defined as Domain 2 in Figure 7, which includes Eastern China and Korea (Figure 8f).

Figure 9 shows monthly mean VCDs of GEMS and TROPOMI averaged in six regions (Figure 7) from August 2020 to December 2022. Values are high in Cambodia and Myanmar, especially in spring, due to biomass burning influences, which are consistently captured by the two products with relatively high correlation coefficients (0.67–0.89). However, GEMS is somewhat inconsistent in regions located in Northeast Asia, such as Korea, North China Plain (NCP), and Yangtze River Delta (YRD), showing low correlation coefficients (0.16–0.40) with TROPOMI, mainly driven by too high GEMS
values in winter. The positive bias of GEMS in winter may be due to high $NO_2$ concentration, which is notable in NCP.

To eliminate artifacts caused by $NO_2$ interference, we empirically corrected glyoxal SCDs using the same linear regression equation derived from the TROPOMI glyoxal retrieval algorithm ($-8.75 \times 10^{12} - 7.01 \times 10^{-3} \times NO_2\ SCD$; Lerot et al., 2021). The corrected GEMS VCDs are depicted in grey lines in Figure 9. Monthly averaged GEMS $NO_2$ SCDs V2.0 in 2021 are used to correct GEMS glyoxal SCDs for all years (2020, 2021, 2022). In NCP, where the $NO_2$
concentration is the highest, mean GEMS $NO_2$ SCDs are $1.57 \times 10^{16}$ molecules cm$^{-2}$ in June 2021 and $2.74 \times 10^{16}$ molecules cm$^{-2}$ in December 2021. The relative differences between GEMS and TROPOMI ($\frac{GEMS-TROPOMI}{TROPOMI}$) glyoxal VCDs in NCP without applying $NO_2$ correction are -5% and 167% in June 2021 and December 2021, respectively. When the $NO_2$ correction is applied, the relative differences are -28% and 49% in June and December 2021, respectively. While the negative bias in the summer worsens to some extent, the positive bias in the winter improves significantly. As a result, the
monthly correlation coefficients improve drastically from 0.16–0.40 to 0.45–0.72 in the regions with high $NO_2$ concentrations, including Korea, NCP, and YRD.

The underestimation of GEMS VCDs compared to TROPOMI in the summer in Northeast Asia could be attributed to the polluted reference spectrum. Although the simulated VCDs are averaged in the same area for the background correction, this does not fully compensate for the reduction in the differential slant column since GEOS-Chem
underestimates glyoxal concentration (Bates and Jacob, 2019; Chan Miller et al., 2017; Silva et al., 2018). This could result in the low seasonal variation of GEMS VCDs, especially in the high latitudes, where the reference sector is somewhat polluted. In Figure 5c, the largest discrepancy depending on the reference sector occurs in 28–40° N, including Korea, NCP, and YRD. Obtaining the reference spectrum from the clean region is very crucial for the GEMS glyoxal product.

## 5 Comparison with MAX-DOAS observations

This section evaluates GEMS VCDs with ground-based Multi-Axis Differential Optical Absorption Spectroscopy (MAX-DOAS) observations (Lerot et al., 2021) at Chiba, Kasuga, Fukue, Phimai, Pantnagar, Haldwani, Seoul, and Xianghe (Figure 7) from August 2020 to December 2021. MAX-DOAS data at Xianghe are operated by BIRA-IASB (Hendrick et al., 2014), and data for the other six stations are operated by CERES (Irie et al., 2011). Each institution uses different instruments and retrieval algorithms (fitting intervals, absorption cross-sections, etc.), leading to the possibility that
instruments from each institution contain distinct systematic biases. Acknowledging this limitation, Lerot et al. (2021)

compared TROPOMI glyoxal VCDs with MAX-DOAS glyoxal VCDs from different institutions. A mean bias of MAX-DOAS glyoxal VCDs from BIRA-IASB ($-0.8 \times 10^{14}$ molecules cm$^{-2}$) is within a range of mean bias of MAX-DOAS glyoxal VCDs from CERES ($-3.5 \times 10^{14} - 0.1 \times 10^{14}$) when compared with TROPOMI glyoxal VCDs. It is uncertain whether there are significant systematic biases between the instruments. Instead, inconsistent biases across the stations could result from the different profiles and aerosol concentrations. The data from Pantnagar and Haldwani were merged and shown in the same subplot, considering their geographical proximity and the lack of temporal overlap; measurements in Pantnagar were made until January 2021, and those in Haldwani were from June 2021. We filtered out MAX-DOAS observations with random uncertainty higher than 30%, primarily from the Pantnagar site. We averaged GEMS VCD pixels within 0.72° from the MAX-DOAS stations for comparison, considering the spatial resolution of GEMS glyoxal data. All the data are hourly and coherently sampled at the same local time. In addition, GEMS VCDs compared with MAX-DOAS VCDs are the values without applying $NO_2$ correction described in Sect. 4.

Figure 10 compares monthly mean GEMS and MAX-DOAS glyoxal VCDs. Both show a reasonable agreement in Northeast Asia (Chiba, Kasuga, Fukue, Seoul, and Xianghe) despite some discrepancies of GEMS being low in summer and high in winter, as shown in the comparison with TROPOMI. In Phimai, GEMS VCDs are slightly lower than MAX-DOAS VCDs but show similar seasonal variations. At the Pantnagar and Haldwani sites, uncertainties are high due to the few MAX-DOAS observations and possible aerosol contamination (Lerot et al., 2021). Underestimation of GEMS VCDs in India is also found in the formaldehyde product, which could be attributed to the longer light path at high viewing zenith angles (Lee et al., 2024).

Figure 11 compares hourly mean GEMS and MAX-DOAS glyoxal VCDs. We find a good agreement between the two datasets regarding glyoxal diurnal variations in Japan (Chiba, Kasuga, and Fukue). Chiba is located near Tokyo and shows elevated concentrations compared to rural sites like Kasuga and Fukue. GEMS underestimates VCDs at the Phimai, Pantnagar, and Haldwani sites, located west of other sites. In Seoul, GEMS and MAX-DOAS VCDs are similar until 14:00 local time and diverge afterward. The continuous increase in the MAX-DOAS observations is not reasonable, though further validation is necessary. GEMS and MAX-DOAS VCDs at Xianghe are consistent except at low solar zenith angles, as fewer samples are available.

**6 Conclusion and discussions**

This study presents the first retrieval of glyoxal columns from a geostationary satellite. To reduce the uncertainty associated with SCD retrieval, we selected optimal settings for the spectral fitting based on sensitivity tests involving the reference spectrum and fitting window. The retrieved SCDs are converted to VCDs using AMFs derived from high-resolution GEOS-Chem simulations. The background correction is the final process of adding column amounts over the

reference sector. The capability of GEMS to observe hourly glyoxal VCDs offers unparalleled temporal resolution, enriching our understanding of VOC emissions and transport.

We compared the retrieved glyoxal VCDs with other satellite and ground-based measurements. GEMS and TROPOMI VCDs generally show similar spatial distribution; however, GEMS VCDs tend to be higher in the north-eastern domain and lower in the south-western domain than TROPOMI VCDs. While monthly variations of GEMS VCDs correlate well with those of TROPOMI and MAX-DOAS VCDs in the Indochinese peninsula regions, variations differ in Northeast Asia. The biases of GEMS may result from the interference with high $NO_2$ concentration, polluted reference spectrum, and high viewing zenith angle.

To address the overestimation in the high $NO_2$ regions, the wavelength dependency of $NO_2$ absorption must be considered. Correction of the glyoxal column with $NO_2$ concentration could improve the consistency of GEMS VCDs with other measurements regarding spatial distribution and temporal variation. This is because the overestimation becomes more pronounced in winter when $NO_2$ concentration is higher. However, subtracting the glyoxal column as a function of $NO_2$ concentration could exacerbate the underestimation of GEMS glyoxal VCDs in summer in Northeast Asia. Therefore, simultaneous work must be performed to resolve the polluted reference spectrum issue. We could consider methods such as filtering pixels representing high glyoxal concentrations simulated from the CTM over the reference sector or utilizing synthetic radiances from the RTM.

The limited field of regard of GEMS poses significant challenges in finding a clean reference sector. While background concentrations in the reference sector are corrected from the simulated concentrations, this is insufficient to resolve bias in GEMS VCDs because the CTM used for background correction underestimates glyoxal VCDs. Enhancing the fidelity of CTMs, particularly in terms of emission and oxidative chemistry of precursor VOCs, is required to mitigate bias in GEMS VCDs across monthly and diurnal variations. While the availability of in-situ glyoxal measurements for reference purposes is limited, the Airborne and Satellite Investigation of Asian Air Quality (ASIA-AQ) campaign presents a promising opportunity. This initiative will comprehensively evaluate observed and simulated glyoxal columns and enhance our understanding of atmospheric processes at scales finer than those resolvable by satellite pixels.

**Data availability.**

The GEMS Level 1C data are available on request from the National Institute of Environmental Research (NIER) – Environmental Satellite Center (ESC). The GEMS Level 2 products are available at https://nesc.nier.go.kr/ko/html/index.do (last access: 28 February 2024). Access to TROPOMI glyoxal tropospheric column data is possible via the GLYRETRO website at https://glyretro.aeronomie.be/index.php/data-menu-item/request-data-test/new-data (last access: 28 February 2024) (Lerot et al., 2021).

**Author contributions.**

ESH, RJP, and HAK designed the study, carried out the analyses, and wrote the manuscript. GTL, SDL, and SS participated in the algorithm development. DWL and HH supported the GEMS instrument. CL, IDS, and TD provided valuable advice on the algorithm development and supplied TROPOMI glyoxal product. FH, HI carried out the MAX-DOAS measurement.

**Competing interests.**

The contact author has declared that none of the authors has any competing interests.

**Special issue statement.**

This article is part of the special issue "GEMS: first year in operation (AMT/ACP inter-journal SI)". It is not associated with a conference.

**Acknowledgements.**

The authors appreciate the GEMS science team and the Environment Satellite Center (ESC) of National Institute of Environmental Research (NIER) for their support in the development of the GEMS glyoxal retrieval algorithm. We would also like to thank the editor and anonymous referees for their thoughtful guidance.

**Financial support.**

This research was supported by a grant from the National Institute of Environmental Research (NIER) (grant no. NIER-2024-04-02-028) and by the Korea Environmental Industry & Technology Institute (KEITI) through the "Climate Change R&D Project for New Climate Regime" (grant no. 2022003560004), funded by the Korea Ministry of Environment (MOE) of the Republic of Korea.

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

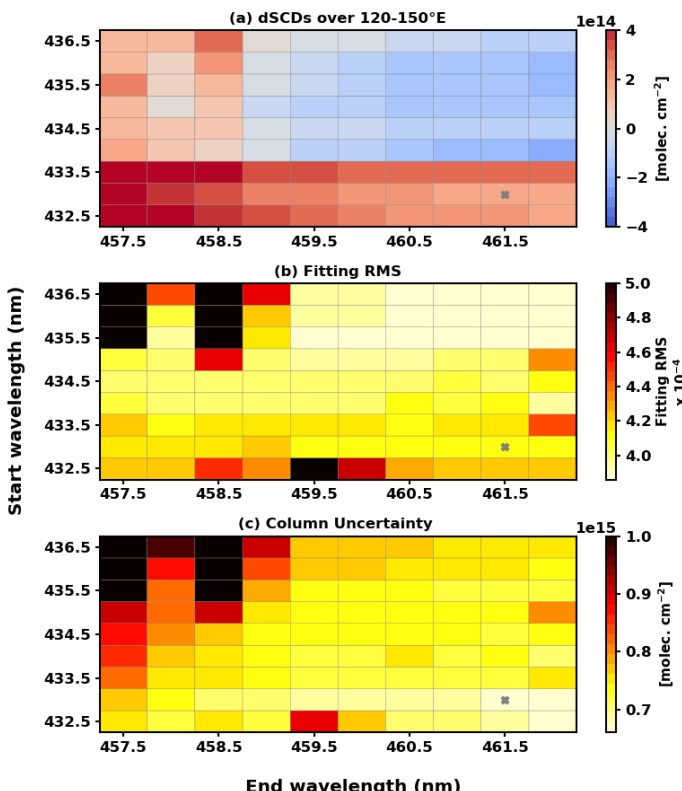

**Figure 1. Metrics used to select an optimal fitting window. Panel a shows dSCDs averaged over 120–150° E, and panels b and c**
**show fitting RMS and column uncertainty averaged over the entire domain. Values are calculated from the retrieval at 04:45 UTC**
**on 17 March 2021.**


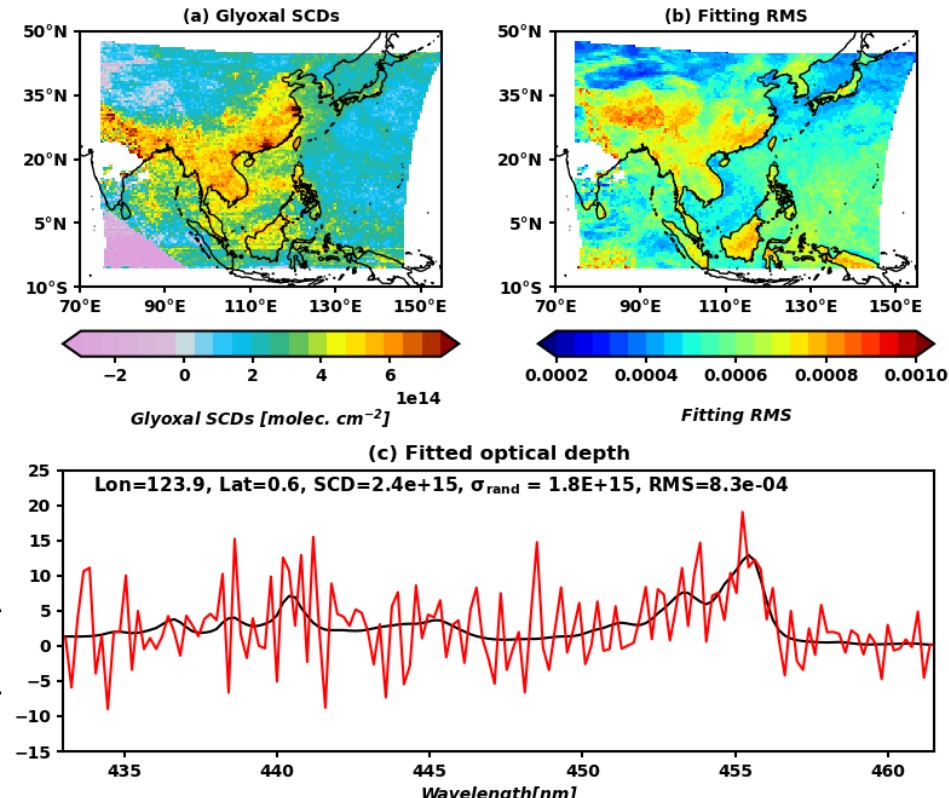

**Figure 2. (a) GEMS glyoxal SCDs and (b) the root-mean-square values from the spectral fit residuals averaged for 02:45–06:45**
**UTC in August 2020. (c) Fitted optical depth (black line) and the sum of optical depth and fitting residual (red line).**

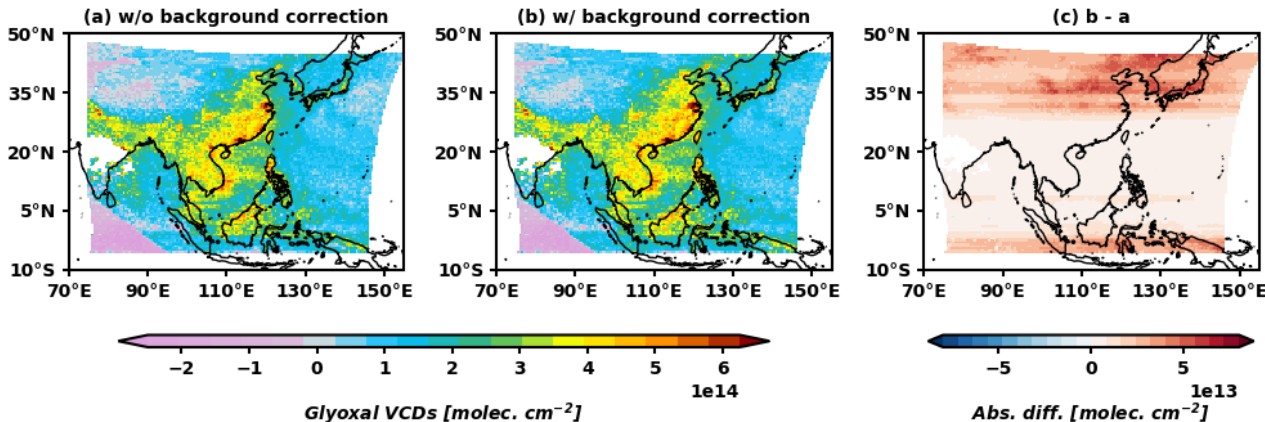

**Figure 3. GEMS glyoxal VCDs retrieved (a) without background correction and (b) with background correction for 02:45–06:45**
**UTC in August 2020. (c) The absolute difference between (a) and (b).**

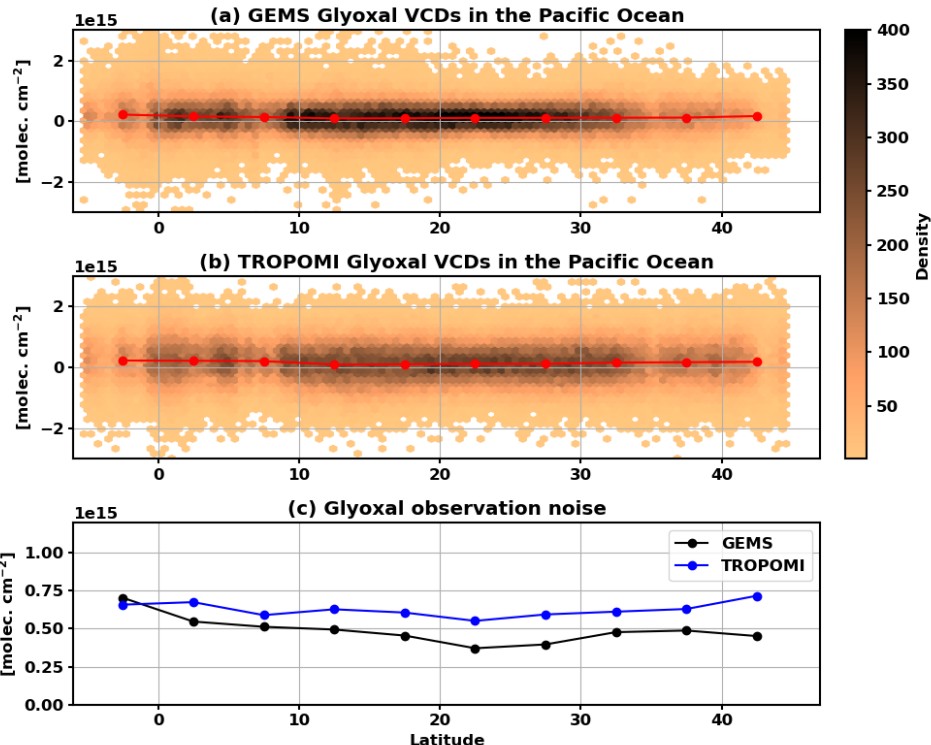

**Figure 4. (a) GEMS and (b) TROPOMI glyoxal VCDs in the ocean over 130–146° E on 3 August 2020. We filtered out pixels over**
**land with cloud cover (cloud fraction > 0.4). The hexagonal heatmap indicates the density of all data, and the red line indicate the data binned in 5° latitude bands. (c) The standard deviation of the binned data for GEMS and TROPOMI.**

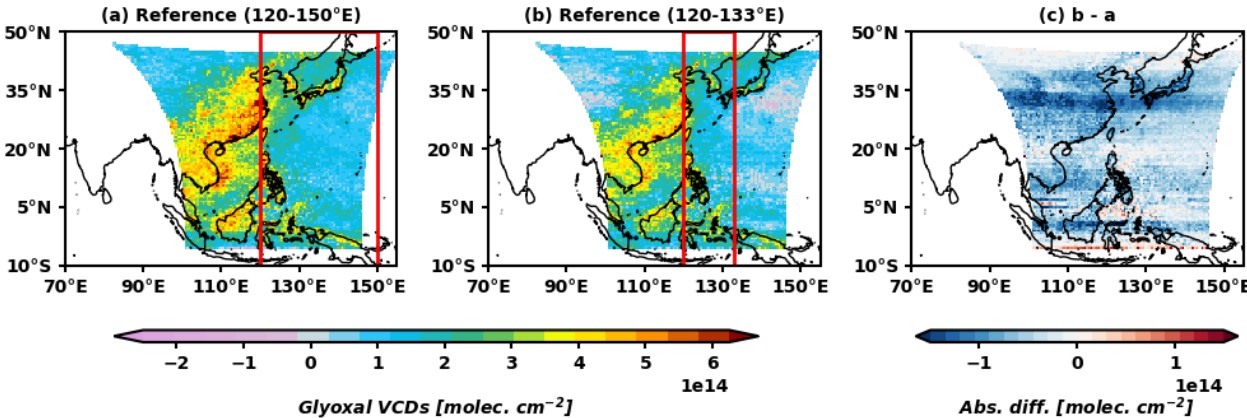

**Figure 5. GEMS glyoxal VCDs averaged for the observations taken as Nominal daily scan for 02:45–06:45 UTC in August 2020. Panel a shows the VCDs retrieved with the reference sector of 120–133° E, and panel b shows those of 120–150° E. The red boxes in panel a and b indicate the reference sector used to retrieve each glyoxal VCDs. (c) The absolute difference between panel a and b.**


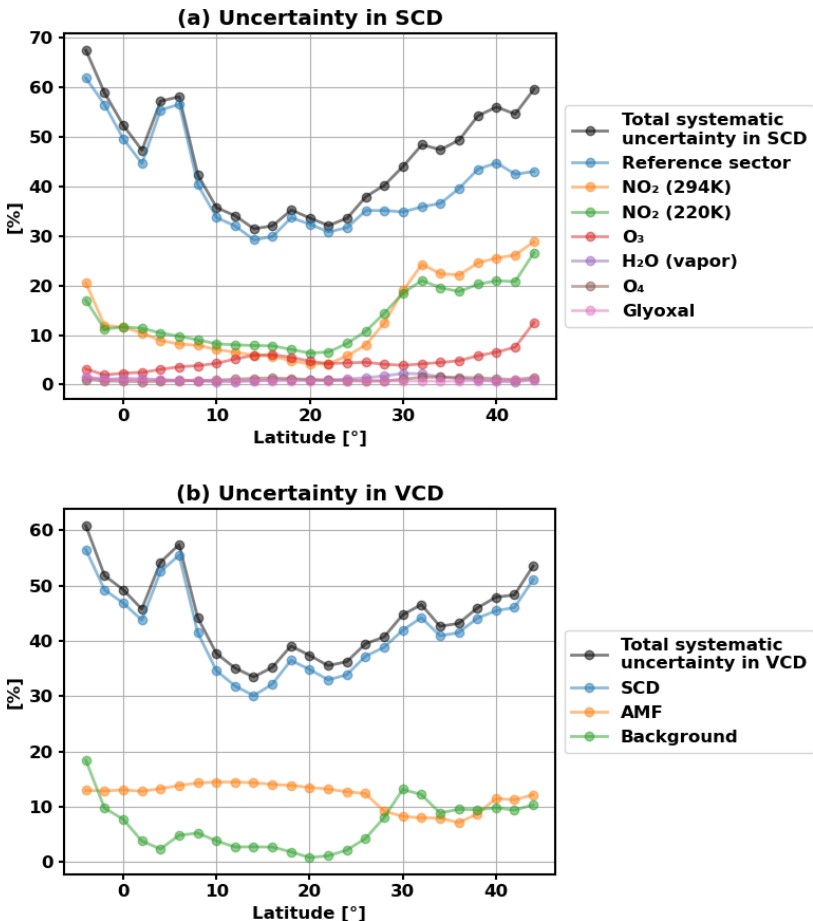


**Figure 6. (a) The systematic uncertainty in SCD averaged at every 2 degrees latitude relative to glyoxal SCD. The total systematic uncertainty in SCD is indicated by the black line, and the contributions from each uncertainty source are indicated by the colored lines. (b) The systematic uncertainty in VCD averaged relative to glyoxal VCD. The total systematic uncertainty in VCD is indicated by the black line, and the contributions from SCD, AMF, and background correction are indicated by the colored lines.**


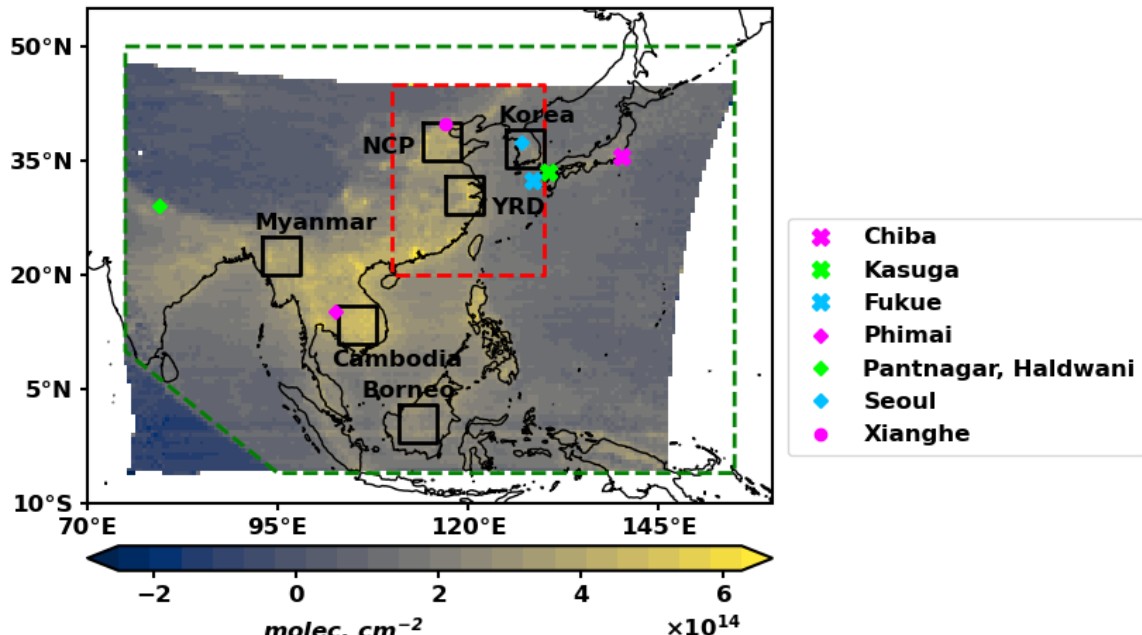

**Figure 7. The green dashed line outlines the area defined as Domain 1, and the red dashed line outlines the area defined as Domain 2 (20–45° N, 110–130° E) in Figure 8. Domain 1 encompasses the area within the green dashed lines, including Domain 2. The black boxes indicate areas where glyoxal VCDs are averaged in Figure 9. Markers indicate the locations of the MAX-DOAS stations in Figure 10 and Figure 11. The colormap in the background represents GEMS glyoxal VCDs averaged from August 2020 to July 2021 at 00:45–07:15 UTC.**


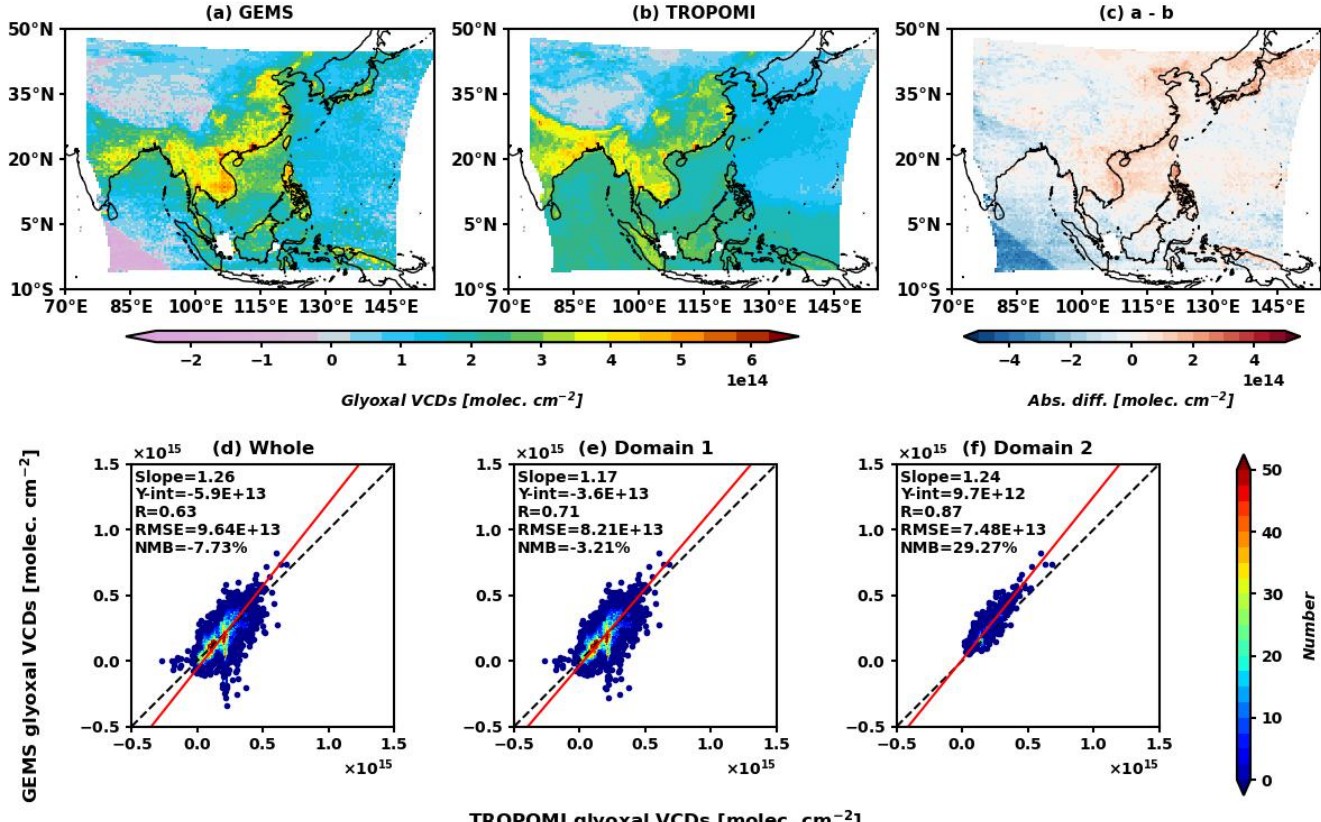

**Figure 8. (a) GEMS and (b) TROPOMI glyoxal VCDs averaged from August 2020 to July 2021 at 11:30–15:30 local time. (c) The absolute difference between GEMS and TROPOMI glyoxal VCDs. Scatter plots comparing GEMS and TROPOMI glyoxal VCDs for the (d) whole domain, (e) Domain 1, and (f) Domain 2 indicated in Figure 7. GEMS VCDs depicted in this plot are the values without applying NO$_2$ correction described in Sect. 4.**

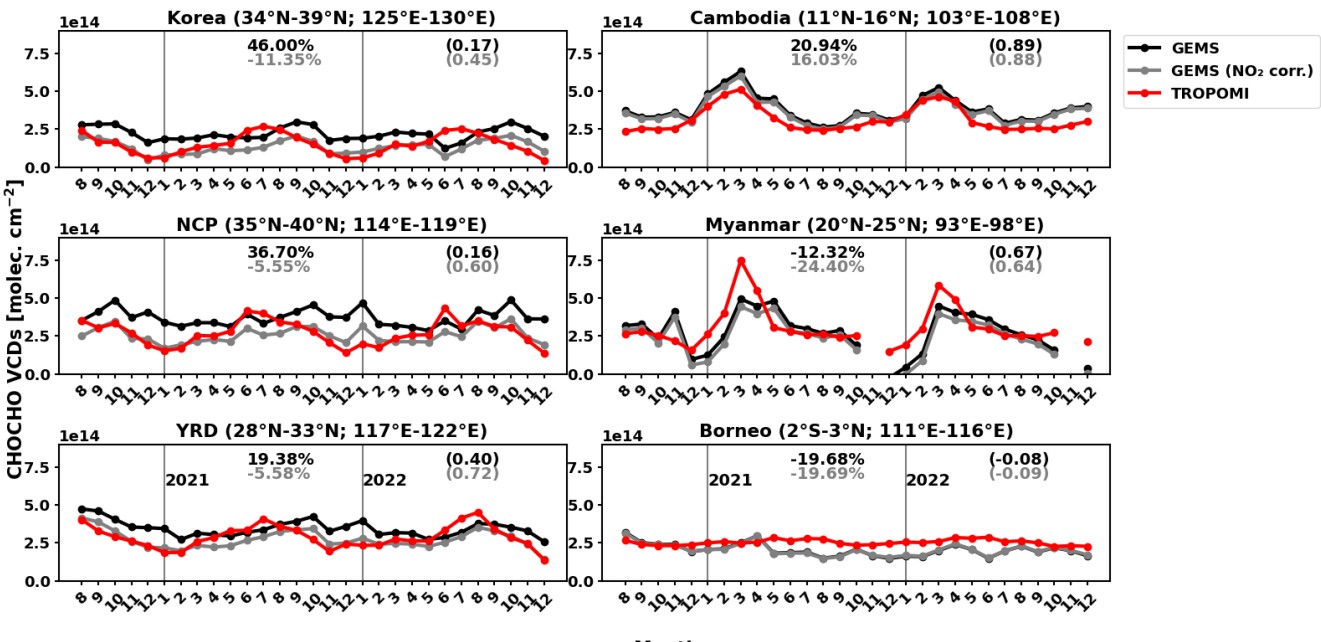

Figure 9. Monthly mean glyoxal VCDs from August 2020 to December 2022 at 11:30–15:30 local time. The black lines represent GEMS VCDs, the grey lines represent GEMS VCDs corrected for NO₂, and the red lines represent TROPOMI VCDs. The numbers on the left denote the normalized mean bias of GEMS VCDs without (black) and with (grey) NO₂ correction relative to TROPOMI VCDs. The numbers in the parentheses in black and grey denote the correlation coefficient of GEMS and TROPOMI VCDs without and with NO₂ correction, respectively.

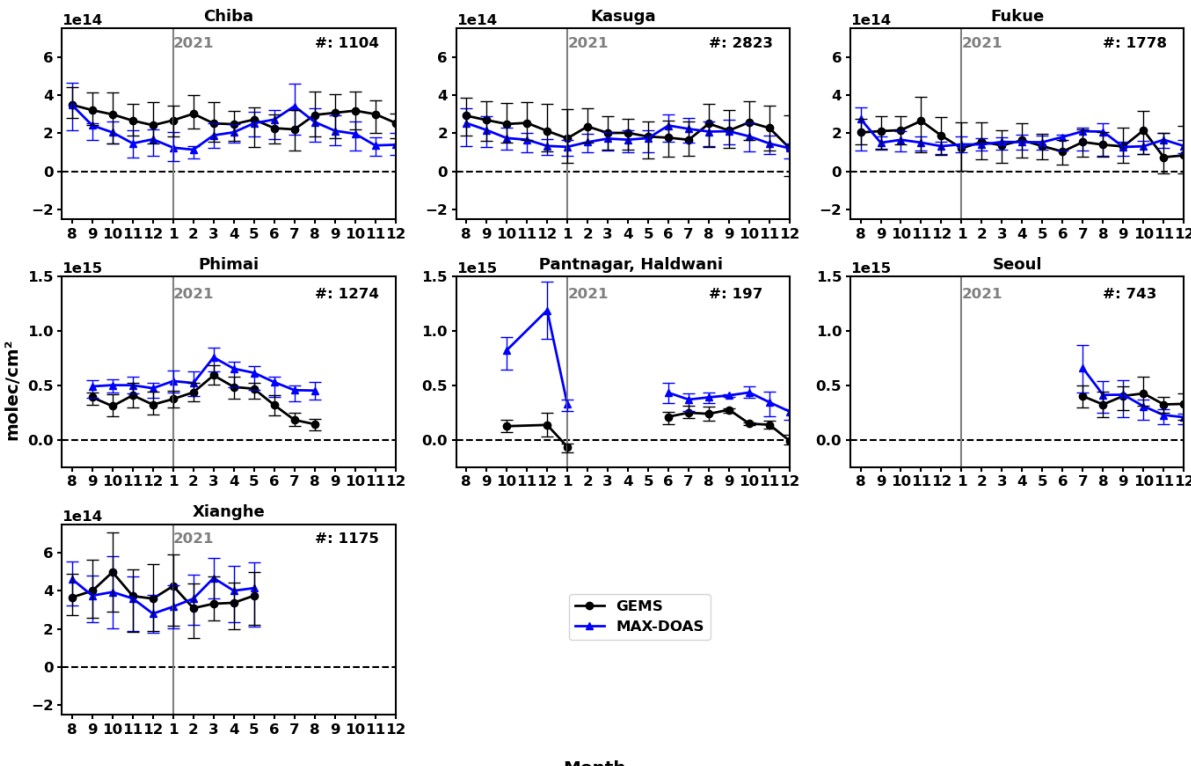

**Figure 10. Monthly mean glyoxal VCDs from GEMS (black line) and MAX-DOAS (blue line) from August 2020 to December 2021. The error bars indicate the 25th and 75th percentiles of hourly averaged VCDs. The numbers on the right denote the number of hourly data co-located at each station.**

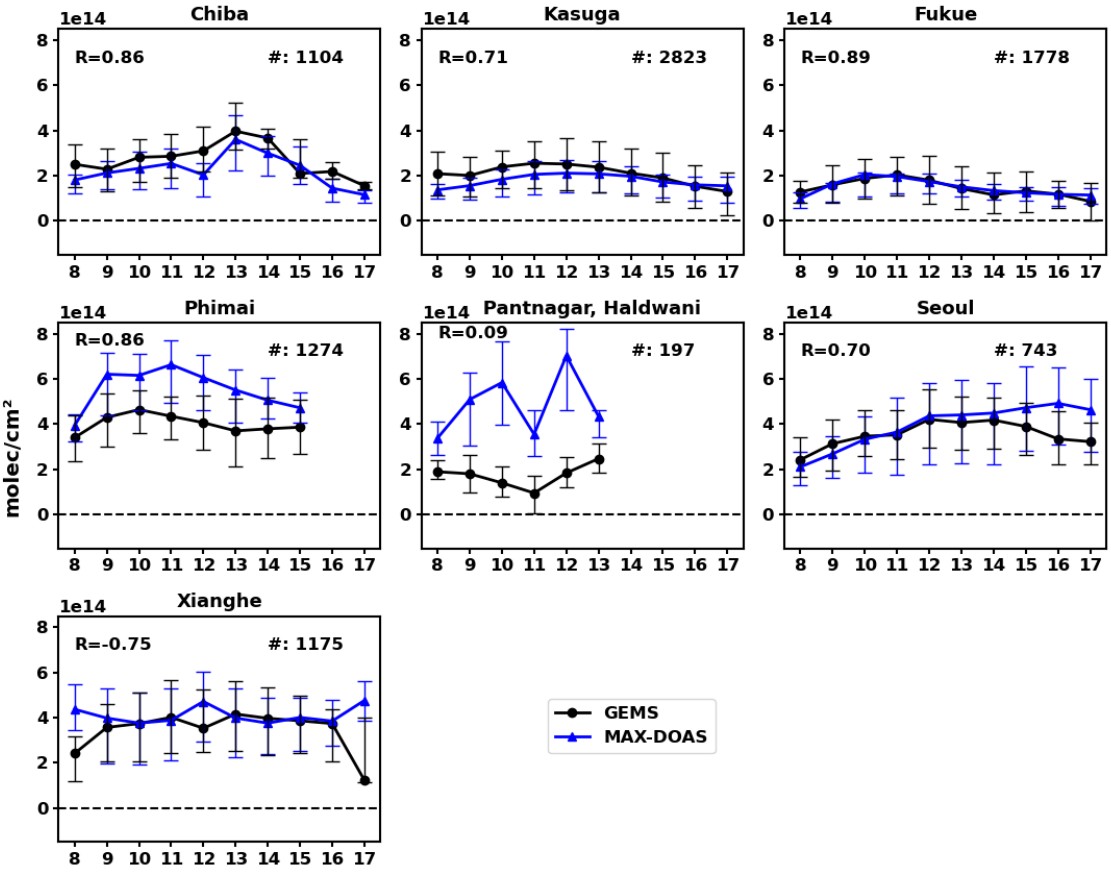

Figure 11. Hourly mean glyoxal VCDs from GEMS (black line) and MAX-DOAS (blue line) from August 2020 to December 2021. The error bars indicate the 25[th] and 75[th] percentiles of hourly averaged VCDs. The numbers on the top left denote the diurnal correlation coefficients between GEMS and MAX-DOAS, and those on the top right denote the number of hourly data co-located at each station.