# Peer review of "First evaluation of the GEMS glyoxal products against TROPOMI and ground-based measurements"

_EGUsphere, 2024_

## Referee Comment (RC2)

**Review:**

**Eunjo Ha et al., First evaluation of the GEMS glyoxal products against TROPOMI and ground-based measurements**

**Summary:**

The authors present retrieved glyoxal columns from the GEMS instrument. Given its short lifetime, measurements from GEO are particularly well-suited to monitoring this trace gas, and the present study is the first of its kind to do this. Glyoxal retrievals from space are challenging due to the gas' low concentrations and to spectral interference from other trace species. The authors have done an excellent job in extracting the glyoxal signal from the data, although the main components of the retrieval algorithm do not differ significantly from approaches in previous studies. I would like to see some additional elaboration on the methods used here, as well as error analyses and studies of sensitivities of the retrieval to assumptions in the algorithm.

The writing in this paper is clear and the manuscript is well organized. Citations are appropriate. With the additions and minor corrections suggested below, I believe it should be published in AMT.

**General comments:**

(1) On page 8 (lines 189-193), the empirical $NO_2$ correction is discussed. When applied, summer differences with TROPOMI become larger, and the winter smaller. In figure 6, or in a separate similar figure, it would be useful to show the GEMS glyoxal with and without the $NO_2$ correction. How does this affect the correlation coefficients? Could differences in correlation lend support the inclusion of the correction?

(2) The VCDs derived relative to spectra from clean reference regions are corrected using GEOS-Chem modeled glyoxal. These model amounts are likely low. Silva et al. suggest the error may be a factor of 3. Have the authors tried increasing the model offsets to counter the effects of these erroneous background values? Perhaps comparisons could be shown.

(3) This paper would benefit from a more comprehensive error analysis, particularly with consideration of the contribution of the major systematic errors, including AMF and background. Glyoxal is a difficult measurement and these errors are likely to be large. Lerot et al. estimated up to 70% error in polluted regions, which I suspect might even be low, given the uncertainties the authors have identified related to background and reference-sector choice.

(4) Were glyoxal amounts in the reference sector estimated at local times comparable to the measurements? If not, perhaps any difference would be negligible. The authors could mention/justify a reason for ignoring this.

**Minor comments and suggested corrections:**

(1) Page 1, Line 20: Without the NO2 correction, GEMS and TROPOMI VCDs are approximately equal in summer. I suggest modifying the wording in the abstract, maybe: "Specifically, with an empirical $NO_2$ correction applied, GEMS VCDs are significantly lower in summer and higher in winter…".

(2) Page 3, Lines 64-67: Please add couple more sentences describing the GEMS instrument, including the native spatial resolution. This will help put the 4 x 4 co-adding in context.

(3) Page 3, Line 75: "…converted to the VCD by dividing by the air mass factor …"

(4) Page 4, Line 100: The fitting window 433.0 – 461.5 nm is approximately the same as used by Lerot et al. and references therein. Did the authors in the present study arrive at this independently? Please clarify and include relevant citations.

(5) Page 6, Line 131: Please explicitly define AMF and $AMF_0$.

(6) Page 9, Line 233: "MAX-DOAS". Also, would it be reasonable to show GEMS vs MAX-DOAS diurnal correlation coefficients for each station?

(7) Page 20, caption figure 4: Please make it clear that Domain 2 is excluded from the surrounding Domain 1 (I assume that is what is meant).

(8) Pages 21-22, captions of figures 5 and 6, (and the body text), please state that the GEMS glyoxal amounts are shown before $NO_2$ correction. As suggested above, I recommend to showing the comparisons with and without the correction

---

## Community Comment (CC1)

**Responses to Referee's Comments**

We are grateful for the referee's valuable and insightful comments. The referee's comments are presented in black, our responses are highlighted in blue, and the revisions made in the manuscript are indicated in red.

**Referee #1:**

Ha et al. introduce the GEMS glyoxal retrieval algorithm and show comparisons against similar observations made from TROPOMI, and a set of MAX-DOAS instruments in the GEMS field of regard. In general they find reasonable agreement with both, however the TROPOMI comparison revealed a bias in GEMS glyoxal at high NO2 concentrations. The correction required in the spectral-fitting algorithm is not currently implemented in the GEMS algorithm.

The paper is well written and will be a useful reference for other researchers who plan to use the product. I recommend publication after the following comments are addressed.

1. L80: Would the spectral fitting not be stable for the retrieval to be performed at native spatial resolution? Aggregation generally causes problems, the most important probably being increased cloud contamination. In general I would have thought the main reason for aggregation is computational expediency (which is fine). I would just have thought that you would get at least equivalent precision/accuracy by aggregating the retrieved glyoxal at native resolution compared to aggregating at L1.

   The reason for co-adding L1 products before the spectral fitting is to ensure stable spectral fitting rather than to enhance computational expediency. We compared the signal-to-noise ratio (SNR) of glyoxal VCDs retrieved at native resolution and co-added resolution. The SNR of glyoxal VCDs retrieved at native resolution and co-added resolution was 0.18 and 0.33, respectively. Figure R1 compares glyoxal VCDs aggregated at L1 and L2 for one scene. Glyoxal VCDs aggregated at L1 and L2 represent similar spatial distribution. However, glyoxal VCDs aggregated at L2 show extreme values along some tracks and in the region where cloud fraction is high. This implies that the retrieval at native resolution is a bit unstable, therefore we use co-added L1 products at the cost of reducing the spatial resolution.

[Figure]

**Figure R1. Comparison of the glyoxal VCDs aggregated at L1 and L2 at 03:45 UTC on 11 March 2021. Panel d is the scatter plot comparing glyoxal VCDs aggregated L1 and L2 with**

**filtering glyoxal VCDs lower than $-1 \times 10^{16}$ molecules cm$^{-2}$ and higher than $2 \times 10^{16}$ molecules cm$^{-2}$.**

2. L85: I know this will be in basically every NO2, HCHO and CHOCHO retrieval paper, but you probably should add equation 6 from Kwon to make the paper self contained.

   I added in the revised manuscript Eq. (6) from Kwon et al. (2019) as Eq. (1) with a brief explanation.

   L85: The modeled radiative transfer equation demonstrates the attenuation of the reference spectrum by gas absorptions based on the Lambert-Beer law (Kwon et al., 2019), and can be expressed as follows:

   $$I(\lambda) = \left[ \left( a I_0(\lambda) + c_r \sigma_r(\lambda) \right) e^{-\sum_i SCD_i \sigma_i(\lambda)} + c_{cm} \sigma_{cm}(\lambda) \right] P_{sc}(\lambda) + P_{bl}(\lambda) \quad (1)$$

   In Eq. (1), $a$ represents an amplification factor applied to the reference spectrum $I_0(\lambda)$; $c_r \sigma_r(\lambda)$ accounts for the contribution of the Ring effect; the term $e^{-\sum_i SCD_i \sigma_i(\lambda)}$ characterizes the attenuation of light by absorbing species i; $c_{cm} \sigma_{cm}(\lambda)$ reflects the influence of the common mode; and $P_{sc}(\lambda)$ and $P_{bl}(\lambda)$ stand for scaling and baseline polynomials, respectively.

3. L90: The spectra region used for the radiance reference may contain significant liquid water absorption, which has caused significant negative biases in retrieved glyoxal, as demonstrated from the previous instruments (GOME-2, OMI) cited in the introduction. If these are used as a radiance reference it may artificially increase glyoxal concentrations over land. Perhaps some of this may be mitigated by the updated Mason and Fry cross section. How does this compare to the previous Pope and Fry liquid water cross section used by the previous studies?

   Thanks for the constructive suggestion. Following your recommendation, we conducted a sensitivity test to retrieve glyoxal VCDs using the liquid water absorption cross-sections from Mason et al. (2016) and Pope and Fry (1997). Figure R2 compares glyoxal VCDs using the two liquid water absorption cross-sections. Glyoxal VCDs increase on land and decrease in the ocean when a liquid water cross-section from Mason et al. (2016) is used. Figure R3 shows the retrieved liquid water dSCDs while retrieving glyoxal VCDs in Figure R2. Liquid water dSCDs retrieved with absorption cross-section from Mason et al. (2016) show lower values on land and higher values in the ocean. Lower liquid water dSCD on land resulted in higher glyoxal VCDs using a liquid water absorption cross-section from Mason et al. (2016). The change in glyoxal VCDs is evident in 0–30° N, which is the latitude range where open oceans exist and liquid water concentration is high. Using absorption cross-section from Mason et al. (2016) increased glyoxal VCDs on land and slightly worsened the overestimation bias of GEMS glyoxal VCDs.

[Figure]

**Figure R2. GEMS glyoxal VCDs retrieved using liquid water absorption cross-sections from (a) Mason et al. (2016) and (b) Pope and Fry (1997) in August 2020. Panel c shows the absolute difference between Panel a and Panel b.**

[Figure]

**Figure R3. Liquid water dSCDs fitted (a) using the same setting used to retrieve GEMS glyoxal V2.0 and (b) using Pope and Fry (1997) liquid water absorption cross-section in August 2020.**

L91: This reference sector contains open oceans where interference of liquid water absorption occurs, which could lead to a high bias of glyoxal VCDs in land. However, we inevitably selected this region as a reference sector since it generally shows low concentrations of glyoxal and other pollutants.

4. L97: The retrieval optimization is mentioned in the intro and conclusion, but not really presented in the text. I think it would be worth adding a figure showing the fit window optimization, and provide more details of the analysis.

We included in the revised manuscript a description of the sensitivity test for the fitting window selection as follows:

L99: We conduct sensitivity tests of fitting window selection to minimize fitting RMS and column uncertainty of the retrieved glyoxal averaged over the entire domain by varying lower and upper wavelengths with 0.5 nm increments. Figure 1 shows the results of our sensitivity tests and our optimal fitting window of 433.0–461.5 nm for glyoxal retrieval. The fitting window of 433.0–461.5 nm was selected considering its low fitting RMS and column uncertainty. However, we find that the differential slant column densities (dSCDs) over the reference sector (120–150° E) retrieved with this fitting window have a positive value, which could result in a high systematic bias, which we discuss below.

[Figure]

**Figure 1. Metrics used to select an optimal fitting window. Panel a shows dSCDs averaged over 120–150°E, and panels b and c show fitting RMS and column uncertainty averaged over the entire domain. Values are calculated from the retrieval at 04:45 UTC on 17 March 2021.**

5.  L117: Is the OMI LER product the most appropriate surface reflectance database for GEMS? Given the different instrument viewing geometries, the equivalent GEMS LER may be significantly different due to BRDF effects, and the OMI LER database spatial resolution is coarse compared to the GEMS pixel size. Are there plans to update this in the future?

It is desirable to use GEMS surface reflectance since it considers BRDF effects and has finer spatial and temporal resolution than OMI LER product. However, GEMS background surface reflectance (BSR) V2.0 has issues such as a distinct bias for different wavelengths compared to TROPOMI directionally-dependent LER (DLER), as well as discontinuity between land and ocean. Figure R4 shows GEMS and OMI surface reflectance and the glyoxal VCDs retrieved with each reflectance. GEMS BSR V2.0 displays much lower values over ocean areas than on land, leading to higher glyoxal VCDs in the ocean than on land. However, advancements have been made in GEMS BSR V3.0, showing more consistent values with OMI LER and TROPOMI DLER. Therefore, we plan to update GEMS glyoxal retrieval to v3.0 using GEMS BSR V3.0 product in the future.

[Figure]

**Figure R4. Surface reflectance values of GEMS BSR V2.0 (a) and OMI LER (b) and glyoxal VCDs retrieved using GEMS BSR V2.0 (c) and OMI LER (d) in March 2022.**

6. L174: Is this paper describing "GEMS glyoxal V2.0"? It probably should be mentioned explicitly somewhere earlier, as it is helpful for users of the product.

I clarified the version information at the beginning of Sect. 2.

L71: The algorithm descriptions in Sect. 2 and the evaluation results in Sect. 3 and Sect. 4 are based on GEMS glyoxal V2.0, which has been the operational product since 2023.

7. L194: Could some of the influence from the polluted background on the reference be eliminated by expanding the time averaging window of the reference radiance, and screening regions that are typically impacted by pollutant outflow? The generally higher retrieved columns in Fig. 7(a) are also what I would expect from the liquid water interference discussed earlier, as the larger reference sector is incorporating more of the open ocean water scenes to the east.

Following your suggestion, we conducted a sensitivity test to retrieve glyoxal VCDs using radiance references by excluding pixels possibly affected by pollution outflow. We used GEOS-Chem results to determine regions that are typically impacted by pollutant outflow. Figure R5 (a) shows regions in the reference sector that we exclude in radiance reference calculation. Panel b shows the absolute difference between glyoxal retrieved with updated radiance references and GEMS glyoxal V2.0. The updated glyoxal VCDs are, in average, 8.8% higher than GEMS glyoxal V2.0 VCDs. We also tested the sensitivity of our retrieval to expanding time averaging of our radiance reference calculation from three to five days. Panel c shows the absolute difference between glyoxal retrieved with five days mean radiance reference and GEMS glyoxal V2.0, which uses three days mean radiance reference. Glyoxal VCDs retrieved with five days mean radiance show uniformly lower values (NMB of -7.1%). Glyoxal VCDs shown in panel d illustrate the effect when the radiance reference is both screened and averaged for five days. The difference is positive north to 30°N and negative in other regions, and the normalized mean bias compared with GEMS V2.0 is 1.0%. Through the sensitivity tests, we confirmed that the current radiance reference that we use is impacted by pollution and this can be partly resolved by excluding regions that show high simulated concentration. Therefore, we plan to find the optimal criteria for screening polluted regions in radiance reference and apply them to GEMS glyoxal V3.0.

Using clean radiance references by excluding polluted regions and incorporating more open ocean as a reference region could contribute to increasing glyoxal VCDs. Two factors, however, results in increases in glyoxal VCDs in different latitude bands such that the first mostly affected regions north to 30°N, while the latter is more important south to 30°N.

[Figure]

Figure R5. Regions in the reference sector that represent column-averaged mixing ratio of glyoxal higher than 1.2 ppt are indicated in red in panel a. The absolute difference between glyoxal VCDs and GEMS glyoxal V2.0 VCDs retrieved using (b) screened radiance reference, (c) five days averaged radiance, and (d) screened and five days average radiance in August 2020.

8. L222: It may be helpful to mention the magnitude of MAX-DOAS instrument-to-instrument biases to aid the interpretation of Fig. 8. Does CERES have some sort of side-by-side intercomparison?

There were no studies that quantified the magnitude of instrument-to-instrument biases in MAX-DOAS glyoxal retrieval as far as I found. However, Lerot et al. (2021) compared TROPOMI glyoxal columns with MAX-DOAS glyoxal columns from different institutions. We included the comparison result of MAX-DOAS and TROPOMI glyoxal VCDs and a notice about the difference in instruments as follows:

L215: Each institution uses different instruments and retrieval algorithms (fitting intervals, absorption cross-sections, etc.), leading to the possibility that instruments from each institution contain distinct systematic biases. Acknowledging this limitation, Lerot et al. (2021) compared TROPOMI glyoxal VCDs with MAX-DOAS glyoxal VCDs from different institutions. Compared with TROPOMI glyoxal VCDs, a mean difference of MAX-DOAS glyoxal VCDs from BIRA-IASB is $-0.8 \times 10^{14}$ molecules cm$^{-2}$, within the range of mean differences of MAX-DOAS glyoxal VCDs from CERES ($-3.5 \times 10^{14} - 0.1 \times 10^{14}$). It is uncertain to confirm if there are significant systematic biases between the instruments. Rather, inconsistent biases across the stations could result from the different profiles and aerosol concentrations.

**References**

Kwon, H. A., Park, R. J., Abad, G. G., Chance, K., Kurosu, T. P., Kim, J., De Smedt, I., Van Roozendael, M., Peters, E., and Burrows, J.: Description of a formaldehyde retrieval algorithm for the Geostationary Environment Monitoring Spectrometer (GEMS), Atmos Meas Tech, 12, 3551–3571, https://doi.org/10.5194/amt-12-3551-2019, 2019.

Lerot, C., Hendrick, F., Van Roozendael, M., Alvarado, L. M. A., Richter, A., De Smedt, I., Theys, N., Vlietinck, J., Yu, H., Van Gent, J., Stavrakou, T., Müller, J. F., Valks, P., Loyola, D., Irie, H., Kumar, V., Wagner, T., Schreier, S. F., Sinha, V., Wang, T., Wang, P., and Retscher, C.: Glyoxal tropospheric column retrievals from TROPOMI -multi-satellite intercomparison and ground-based validation, Atmos Meas Tech, 14, 7775−7807, https://doi.org/10.5194/amt-14-7775-2021, 2021.

Mason, J. D., Cone, M. T., and Fry, E. S.: Ultraviolet (250–550 nm) absorption spectrum of pure water, Appl Opt, 55, 7163, https://doi.org/10.1364/ao.55.007163, 2016.

Pope, R. M., and Fry, E. S.: Absorption spectrum (380–700 nm) of pure water, II. Integrating cavity measurements, Appl Opt, 36, 8710–8723, https://doi.org/10.1364/AO.36.008710, 1997.

---

## Author Comment (AC1)

**Responses to Referee's Comments**

We are grateful for the referee's valuable and insightful comments. The referee's comments are presented in black, our responses are highlighted in blue, and the revisions made in the manuscript are indicated in red.

**Referee #2:**

The authors present retrieved glyoxal columns from the GEMS instrument. Given its short lifetime, measurements from GEO are particularly well-suited to monitoring this trace gas, and the present study is the first of its kind to do this. Glyoxal retrievals from space are challenging due to the gas' low concentrations and to spectral interference from other trace species. The authors have done an excellent job in extracting the glyoxal signal from the data, although the main components of the retrieval algorithm do not differ significantly from approaches in previous studies. I would like to see some additional elaboration on the methods used here, as well as error analyses and studies of sensitivities of the retrieval to assumptions in the algorithm.

The writing in this paper is clear and the manuscript is well organized. Citations are appropriate. With the additions and minor corrections suggested below, I believe it should be published in AMT.

**General comments:**

1. On page 8 (lines 189-193), the empirical NO2 correction is discussed. When applied, summer differences with TROPOMI become larger, and the winter smaller. In figure 6, or in a separate similar figure, it would be useful to show the GEMS glyoxal with and without the NO2 correction. How does this affect the correlation coefficients? Could differences in correlation lend support the inclusion of the correction?

In the same figure, we depicted the monthly variation of GEMS VCDs with and without applying the $NO_2$ correction. When the correction is applied, GEMS glyoxal VCDs decrease in larger amounts in winter than in summer in the regions with high $NO_2$ concentrations, such as Korea, NCP, and YRD. This substantially increases the correlation coefficients, improving from 0.16–0.40 to 0.45–0.72 in these high $NO_2$ regions. Therefore, it is highly likely that $NO_2$ inhibits accurate representation of monthly variation, necessitating the correction for $NO_2$.

L281: To eliminate artifacts caused by $NO_2$ interference, we empirically corrected glyoxal SCDs using the same linear regression equation derived from the TROPOMI glyoxal retrieval algorithm ($-8.75 \times 10^{12} - 7.01 \times 10^{-3} \times NO_2\ SCD$; Lerot et al., 2021). The corrected GEMS VCDs are depicted in grey lines in Figure 9. Monthly averaged GEMS $NO_2$ SCDs V2.0 in 2021 are used to correct GEMS glyoxal SCDs for all years (2020, 2021, 2022). In NCP, where the $NO_2$ concentration is the highest, mean GEMS $NO_2$ SCDs are $1.57 \times 10^{16}$ molecules cm$^{-2}$ in June 2021 and $2.74 \times 10^{16}$ molecules cm$^{-2}$ in December 2021. The relative differences between GEMS and TROPOMI ($\frac{GEMS-TROPOMI}{TROPOMI}$) glyoxal VCDs in NCP without applying $NO_2$ correction are -5% and 167% in June 2021 and December 2021, respectively. When the $NO_2$ correction is applied, the relative differences are -28% and 49% in June and December 2021, respectively. While the negative bias in the summer worsens to some extent, the positive bias in the winter improves significantly. As a result, the monthly correlation coefficients improve drastically from 0.16–0.40 to 0.45–0.72 in the regions with high $NO_2$ concentrations, including Korea, NCP, and YRD.

[Figure]

**Figure 9. Monthly mean glyoxal VCDs from August 2020 to December 2022 at 11:30–15:30 local time. The black lines represent GEMS VCDs, the grey lines represent GEMS VCDs corrected for NO₂, and the red lines represent TROPOMI VCDs. The numbers on the left denote the normalized mean bias of GEMS VCDs without (black) and with (grey) NO₂ correction relative to TROPOMI VCDs. The numbers in the parentheses in black and grey denote the correlation coefficient of GEMS and TROPOMI VCDs without and with NO₂ correction, respectively.**

2. The VCDs derived relative to spectra from clean reference regions are corrected using GEOS-Chem modeled glyoxal. These model amounts are likely low. Silva et al. suggest the error may be a factor of 3. Have the authors tried increasing the model offsets to counter the effects of these erroneous background values? Perhaps comparisons could be shown.

To confirm the effect of the increase in background values, we tripled the background used in GEMS glyoxal V2.0, as suggested by Silva et al. (2018). The resulting glyoxal VCDs are shown as the blue lines in Figure R1. Background values have minimal impacts on VCDs within the latitude range of 3° N to 25° N, which includes Cambodia and Myanmar. The reference sector in this latitude range is mostly occupied by the ocean, thereby representing low simulated glyoxal VCDs. In contrast, there are considerable changes in VCDs in other latitude ranges, as shown in the monthly variation of Korea, NCP, YRD, and Borneo. GEMS VCDs with NO₂ correction and using background values from GEMS glyoxal V2.0 show underestimation in summer. When the background is increased threefold, GEMS VCDs nearly match TROPOMI VCDs in summer but exceed TROPOMI VCDs in other seasons.

[Figure]

**Figure R1. Monthly mean glyoxal VCDs from August 2020 to December 2022 at 11:30–15:30 local time. The red lines represent TROPOMI VCDs, the grey lines represent GEMS VCDs corrected for NO₂, and the blue lines represent GEMS VCDs corrected for NO₂ and using three times larger background values.**

3.  This paper would benefit from a more comprehensive error analysis, particularly with consideration of the contribution of the major systematic errors, including AMF and background. Glyoxal is a difficult measurement and these errors are likely to be large. Lerot et al. estimated up to 70% error in polluted regions, which I suspect might even be low, given the uncertainties the authors have identified related to background and reference-sector choice.

We added Sect. 3 discussing the uncertainty of each retrieval step.

L163: **3 Uncertainty analysis**

In this section, we examine the uncertainty of each retrieval step. Assuming that each retrieval step is uncorrelated, the total uncertainty in the retrieval can be expressed as the sum of the uncertainties of each retrieval step (Boersma et al., 2004; Kwon et al., 2019; De Smedt et al., 2018):

$$\sigma_{VCD}{}^2 = \left(\frac{\partial VCD}{\partial dSCD}\sigma_{SCD}\right)^2 + \left(\frac{\partial VCD}{\partial AMF}\sigma_{AMF}\right)^2 + \left(\frac{\partial VCD}{\partial VCD_m}\sigma_{VCD_m}\right)^2 + \left(\frac{\partial VCD}{\partial AMF_0}\sigma_{AMF_0}\right)^2, \quad (4)$$

where $\sigma$ denotes uncertainty in each parameter. Using the relationship shown in Eq. (3), we can derive the sensitivity of VCDs to each parameter, resulting in the conversion of Eq. (4) as follows:

$$\sigma_{VCD}{}^2 = \frac{1}{AMF^2}\sigma_{SCD}{}^2 + \frac{SCD^2}{AMF^4}\sigma_{AMF}{}^2 + \frac{AMF_0{}^2}{AMF^2}\sigma_{VCD_m}{}^2 + \frac{VCD_m{}^2}{AMF^2}\sigma_{AMF_0}{}^2. \quad (5)$$

Uncertainty in each parameter includes both random and systematic components. Unlike systematic uncertainty, random uncertainty decreases for a spatial or temporal averaging in the ratio of $\frac{1}{\sqrt{N}}$, where N is the number of pixels averaged. In this study, we only consider random uncertainty in slant columns as random uncertainty in AMF and $VCD_m$ are difficult to separate from the systematic uncertainty in practice.

**3.1 Uncertainties in slant columns**

[revised manuscript text omitted]

4. Were glyoxal amounts in the reference sector estimated at local times comparable to the measurements? If not, perhaps any difference would be negligible. The authors could mention/justify a reason for ignoring this.

MAX-DOAS stations at Chiba, Kasuga, Fukue, and Seoul are included in the reference sector (120–150° E). The diurnal variations of GEMS VCDs in these stations are consistent with MAX-DOAS VCDs, showing high correlation coefficients (0.70–0.89) and low normalized mean biases (-9–20%). However, these regions are more polluted than the rest of the reference sector, and the ratios of the background values over VCDs are larger in the rest of the reference sector. Evaluating GEMS VCDs in the pristine region of the reference sector using independent measurements would validate the use of reference sector and background values. Unfortunately, there are few in situ measurements available for comparison with GEMS glyoxal over the ocean during the GEMS observation period.

**Minor comments and suggested corrections:**

1. Page 1, Line 20: Without the NO2 correction, GEMS and TROPOMI VCDs are

approximately equal in summer. I suggest modifying the wording in the abstract, maybe: "Specifically, with an empirical NO2 correction applied, GEMS VCDs are significantly lower in summer and higher in winter…".

We revised the abstract as follows:

L20: Specifically, GEMS VCDs are higher in the winter and either lower or comparable to TROPOMI and MAX-DOAS VCDs in the summer across Northeast Asia. We attributed the discrepancies in the monthly variation to a polluted reference spectrum and high $NO_2$ concentrations. When we correct GEMS glyoxal VCDs as a function of $NO_2$ SCDs, the monthly correlation coefficients substantially increase from 0.16–0.40 to 0.45–0.72 in high $NO_2$ regions.

2. Page 3, Lines 64-67: Please add couple more sentences describing the GEMS instrument, including the native spatial resolution. This will help put the 4 x 4 co-adding in context.

We added the following sentence:

L83: The native spatial resolution of GEMS is $3.5 \times 8 \text{ km}^2$, with nitrogen dioxide, ozone, formaldehyde, and aerosol products retrieved at this resolution. For the weaker absorbers, we use co-added products, including radiance, irradiance, surface reflectance, and cloud products. Specifically, we co-add products with 4 (2×2) and 16 (4×4) GEMS pixels to retrieve sulfur dioxide and glyoxal, respectively, to enhance the signal-to-noise ratio.

3. Page 3, Line 75: "…converted to the VCD by dividing by the air mass factor …"

We revised the phrase in the manuscript according to your correction.

4. Page 4, Line 100: The fitting window 433.0 – 461.5 nm is approximately the same as used by Lerot et al. and references therein. Did the authors in the present study arrive at this independently? Please clarify and include relevant citations.

We determined the optimal fitting window through sensitivity tests on various fitting windows. A description and figure of the sensitivity test results have been added.

L112: We conduct sensitivity tests of fitting window selection to minimize fitting RMS and column uncertainty of the retrieved glyoxal averaged over the entire domain by varying lower and upper wavelengths with 0.5 nm increments. Figure 1 shows the results of our sensitivity tests and our optimal fitting window of 433.0–461.5 nm for glyoxal retrieval. The fitting window of 433.0–461.5 nm was selected considering its low fitting RMS and column uncertainty. However, we find that the differential slant column densities (dSCDs) over the reference sector (120–150° E) retrieved with this fitting window have a positive value, which could result in a high systematic bias, which we discuss below.

[Figure]

**Figure 1. Metrics used to select an optimal fitting window. Panel a shows dSCDs averaged over 120–150° E, and panels b and c show fitting RMS and column uncertainty averaged over the entire domain. Values are calculated from the retrieval at 04:45 UTC on 17 March 2021.**

5.  Page 6, Line 131: Please explicitly define AMF and AMF0.

    L148: $AMF(i,j)$ indicates the AMF at the $i$th cross-track (north-south direction) and $j$th along-track (east-west direction) positions, and $AMF_0(lat)$ denotes the AMF over the reference sector.

6.  Page 9, Line 233: "MAX-DOAS". Also, would it be reasonable to show GEMS vs MAX-DOAS diurnal correlation coefficients for each station?

    We corrected the typo and added the diurnal correlation coefficients to Figure 11.

[Figure]

**Figure 11. Hourly mean glyoxal VCDs from GEMS (black line) and MAX-DOAS (blue line) from August 2020 to December 2021. The error bars indicate the 25th and 75th percentiles of hourly averaged VCDs. The numbers on the top left denote the diurnal correlation coefficients between GEMS and MAX-DOAS, and those on the top right denote the number of hourly data co-located at each station.**

7. Page 20, caption figure 4: Please make it clear that Domain 2 is excluded from the surrounding Domain 1 (I assume that is what is meant).

   We intended Domain 1 to include Domain 2, so we corrected and clarified the caption as follows:

   L585: The green dashed line outlines the area defined as Domain 1, and the red dashed line outlines the area defined as Domain 2 (20–45° N, 110–130° E) in Figure 8. Domain 1 encompasses the area within the green dashed lines, including Domain 2.

8. Pages 21-22, captions of figures 5 and 6, (and the body text), please state that the GEMS glyoxal amounts are shown before NO2 correction. As suggested above, I recommend to showing the comparisons with and without the correction

   We showed comparisons with and without the correction in Figure 9. Also, we stated that the GEMS glyoxal amounts are shown before the NO$_2$ correction as follows:

   L315 : In addition, GEMS VCDs compared with MAX-DOAS VCDs are the values without applying NO$_2$ correction described in Sect. 4.
   L596: GEMS VCDs depicted in this plot are the values without applying NO$_2$ correction described in Sect. 4.